# A Survey of the Knowledge, Attitudes and Practices of a Sample of Albanian Medical Students in Relation to Occupational Exposure to Biological Agents

**DOI:** 10.3390/diseases13010011

**Published:** 2025-01-10

**Authors:** Lorenzo Ippoliti, Luca Coppeta, Ersilia Buonomo, Giuseppina Somma, Giuseppe Bizzarro, Cristiana Ferrari, Andrea Mazza, Agostino Paolino, Claudia Salvi, Vittorio Caputi, Antonio Pietroiusti, Andrea Magrini

**Affiliations:** 1Department of Biomedicine and Prevention, University of Rome Tor Vergata, 00133 Rome, Italy; 2Faculty of Medicine, Saint Camillus International University of Health Sciences, 00131 Rome, Italy; 3Faculty of Medicine, University “Our Lady of Good Counsel”, 1000 Tirana, Albania

**Keywords:** risk perception, occupational blood-borne pathogen exposure, medical student knowledge, attitudes

## Abstract

(1) Background: Exposure to blood carries the risk of transmission of many infectious diseases. Healthcare workers (HCWs), including hospital-based medical students, face high and often under-reported rates of exposure to needlestick and sharps injuries. Previous studies have shown that students’ knowledge of infection control varies, highlighting the importance of pre-placement training. This study aims to assess knowledge, attitudes and practices regarding these risks in a population of medical students from Albania. (2) Methods: A validated questionnaire was administered to 134 medical students in an Italian hospital in May 2023. It assessed HBV vaccination status, adherence to infection control practices, knowledge of pathogen transmission, exposure incidents and attitudes towards infected patients. Three additional questions addressed air-borne transmission of tuberculosis and vaccination recommendations for healthcare workers. (3) Results: Most students (64%) reported being aware of occupational exposure risks. While 93% and 87%, respectively, recognised HIV and HBV as blood-borne pathogens, fewer recognised Treponema pallidum (44%). Awareness of post-exposure prophylaxis for HIV was high (85%), but although 75% reported having received training, only 45% felt it was adequate. Statistical analysis revealed an association between knowledge of infection control, awareness of pathogen transmission and understanding of the importance of vaccination. (4) Conclusions: Our study highlights gaps in medical students’ knowledge of occupational infections and highlights the need for improved pre-clerkship education. Improved education could reduce anxiety, ethical issues and misconceptions and promote safer healthcare practices.

## 1. Introduction

Exposure to infected blood carries the risk of transmission of blood-borne pathogens, including human immunodeficiency virus (HIV), hepatitis B virus (HBV), hepatitis C virus (HCV) and Treponema pallidum (TP) [1]. Healthcare workers (HCWs) may experience occupational blood exposures when they encounter a needlestick or sustain a percutaneous injury due to a contaminated sharp instrument. Awareness is crucial as exposure to these substances can lead to the contraction and spread of these serious infections [2,3].

The prevalence of HBV is estimated to be around 0.9% and that of HCV is about 1.1% in the European Union, with an estimated total of 4.7 million chronic HBV cases and 5.6 million HCV cases [4]. In 2022, 110,486 HIV diagnoses were reported in Europe, corresponding to 12.4 HIV diagnoses per 100,000 inhabitants for the region as a whole, a slight increase compared with the rate in 2021 (11.9 per 100,000 inhabitants) [5].

Regarding the situation in Albania, despite decreasing HBsAg prevalence estimates, the country remains highly endemic for HBV, with an HBsAg prevalence rate of over 8% [6]. However, recent data on HBV prevalence are limited. The prevalence of HCV infection in the general population is around 1% [7]. Albania is considered a low-prevalence country for HIV, with an estimated rate of 0.04% in the general population [8]. The incidence of tuberculosis has improved over the years, decreasing from 22 cases per 100,000 people in 2000 to 15 cases per 100,000 in 2023 [9]. Despite the Ministry of Health’s approval of a National Measles Elimination Plan in 2000 and minimal cases reported over the past two decades [10], measles resurfaced in 2018–2019 with hospitalisations with different clinical manifestations, highlighting its status as a re-emerging infectious disease in the country [11].

Hospital placements are an obvious risk for medical students. The incidence of accidents due to blood exposure among hospital medical students is high and probably underestimated by official statistics due to the low declaration rate [12,13,14]. It is crucial that medical students are fully aware of the risks associated with their placement.

Needlestick injuries (NSIs) are a major occupational hazard for healthcare workers (HCWs) worldwide, including in Central and Eastern Europe. A systematic review and meta-analysis estimated that 44.5% of HCWs worldwide experience at least one NSI per year [15]. Data on NSIs in Albania are limited. In Bosnia and Herzegovina, a hospital-based study found that 63.3% of HCWs reported exposure to blood and body fluids during their working life, with NSIs (66.1%) being the most common cause, followed by skin contact (12.1%) and cuts from sharp objects (11.3%) [16]. In Turkey, a study of nursing students found that needles caused 54.0% of injuries, mainly during intravenous or intramuscular injections (60.0%). Notably, 31.7% of students did not wear gloves when injured and 68.3% of NSIs were not reported [17].

Although glove use reduces the risk of sharps injuries, compliance among HCWs is variable and influenced by risk perception and workplace culture [18]. In agreement with other European studies, our results highlight that safety-engineered devices significantly reduce the risk of percutaneous exposure among HCWs [19].

Several studies have explored the knowledge, attitudes and practices of health profession students, reporting conflicting results. In some cases, students seem to have important knowledge about infection control [20], while in others they seem not to be fully aware of the risks posed by the work experience and post-exposure measures [21,22], and some studies have reported a low level of needlestick practice [23]. Other studies have shown that healthcare students are aware of the risks of unintentional exposure to HBV [24], although good pre-placement training on the subject is undoubtedly crucial [25]. On the basis of these data, we felt it was essential to investigate the level of knowledge, attitudes and practices regarding exposure to blood-borne and air-borne pathogens in a population of medical students prior to the beginning of their internship at our hospital; in particular, our intention was to assess these aspects in a population of students from the Balkan region.

We conducted this study to investigate medical students’ knowledge of pathogen transmission, prevention and the management of occupational exposure. We also investigated the frequency and details of medical students’ occupational exposure and assessed medical students’ attitudes towards the impact of pathogen risk on future careers and their attitudes towards infected patients.

## 2. Materials and Methods

The survey was conducted in an Italian hospital where an Albanian study group (*n* = 134) was on placement during the month of May 2023. The questionnaire used was validated by a previous study on the same type of operators [25]. The survey included questions on various aspects of the students, such as their general characteristics, whether they had received the HBV vaccine, their adherence to infection control procedures during normal clinical activities and their understanding of blood-borne pathogen transmission and occupational exposure. The questionnaire also explored exposure incidents, students’ attitudes towards training, their professional and practical experience and their willingness to care for HIV/HBV patients. In addition to the validated questionnaire, we included 3 additional items about knowledge of air-borne transmission of Mycobacterium Tuberculosis (MT) and about influenza and measles vaccinations, specifically asking whether they were recommended for HCWs. The survey was distributed to students who came to the occupational health department for a consultation prior to their work placement (see Appendix A). All students participated anonymously and gave informed consent.

Subject characteristics are reported as numbers and percentages, depending on the type of variable. A chi-square test was used to analyse the data. *p* < 0.05 was considered statistically significant. All analyses were performed with SPSS version 25.0 for Windows.

## 3. Results

A total of 134 questionnaires were analysed. Although 138 questionnaires were collected (100% response rate), 4 questionnaires were excluded from the analysis because they were incomplete. The characteristics of the study population are shown in Table 1.

### 3.1. Knowledge

The majority of respondents (64%, n = 86) stated that they were fully aware of the problem of infectious occupational exposure. In addition, 84.3% (n = 113) declared that they have a good understanding of proper hand hygiene. Regarding blood-borne pathogens, the majority of students were aware that HIV (93% n = 125) and HBV (87% n = 117) are transmitted through blood, but less than half (45% = 59) recognised that TP is transmitted in this way (Figure 1).

In the section on post-exposure procedures, the majority of participants were aware of HIV prophylaxis regimens (85%, n = 114), but only 55% (n = 74) were aware of the possibility of using the HBV vaccine in some post-exposure prophylaxis regimens. In addition, according to 27% (n = 36) of participants, the HCV vaccine can be used in post-exposure prophylaxis.

Regarding the three additional items, 94% (n = 126) of students answered that MT can be transmitted by respiratory droplets; 74.6% (n = 100) and 64.2% (n = 86) of participants were aware that influenza and measles vaccinations are strongly recommended for HCWs, respectively.

### 3.2. Professional Exposures

Of the total, only two participants reported a biological injury. In the first case, the injury occurred in the 6 months prior to the questionnaire, due to exposure to abrasions during a manoeuvre involving the use of sharps (surgery). In the second case, the injury occurred more than a year before, due to a skin wound from a needle during a blood test. In both cases, the students reported that they had all the necessary personal protective equipment and that they had not received any post-exposure treatment because the source patients were negative for infectious diseases.

### 3.3. Attitudes

All students agreed that it was very important to have in-depth knowledge of the pathogens they might be exposed to in their work, but only 74.6% (n = 100) declared that they had received education on the matter, and only 44.8% (n = 60) stated that it had been sufficient.

In terms of future work choices, 47.0% (n = 63) said that the risk of exposure to biological agents would influence their careers. In addition, 79.8% (n = 107) said that this risk of professional exposure will affect the way they work and their subsequent practices on infection control.

In addition, five students (4.7%) reported that they refused to treat infectious patients for fear of transmission.

Nevertheless, regarding common infection control practices, as shown in Figure 2, the students surveyed do not have a full understanding of the regulations.

### 3.4. Statistical Analysis

A chi-squared test was performed between items 34 and 35 (vaccination recommendations) and items 2, 3, 4, 14, 24, 25, 26, 29, 30 and 33; see Table 2 and Table 3.

A statistically significant association was found for both items (influenza and measles vaccination) with knowledge of parenteral transmission of Treponema pallidum and with knowledge of aerogenic transmission (droplets) of MT.

Another statistical analysis (chi-square test) was also carried out to assess the correlation between item 10C and the previously analysed items (2, 3, 4, 14, 24, 25, 26, 29, 30 and 33); see Table 4.

Again, a statistically significant association was found between knowledge of parenteral transmission of Treponema pallidum and items 25, 26 and 30 (“Do you think that training on occupational exposure for medical students is sufficient?”, “After learning about the risks of occupational exposure to biological agents do you think the risk will influence your future career choices?” and “Have you ever refused to treat an infectious patient for fear of transmission?”).

## 4. Discussion

The problem of biohazard exposure among students in health education courses is a topical issue that requires attention. It is crucial to provide adequate education and training to prospective students before they begin their journey in hospital facilities. Medical students seem to be less knowledgeable than other HCWs about healthcare-associated infections [26]. Undergraduate years are the appropriate time for acquiring the necessary knowledge and skills in this area [27]. In addition, training in infection control measures improves the compliance of HCWs with the standard precautions [28]. Such measures are essential to protect patients, HCWs and students [29].

Our study was conducted on a sample of 138 medical students from an Albanian university who were about to start their internship in our hospital. The analysis was possible because the students were visited in the occupational medicine department before starting their internship and all students were given the evaluation questionnaire. Unfortunately, the results showed that a significant percentage of the students had a significant lack of knowledge about healthcare-associated infections.

The results show the importance of improving pre-service training programmes and ensuring that the medical course includes sufficient lectures on occupational hazards. We could suggest the possibility of including pre-clinical practice examinations in order to highlight the deficiencies of the students prior to the placement.

Our analysis revealed a statistically significant association between knowledge of influenza and measles vaccination recommendations and participants’ understanding of the modes of transmission of both Treponema pallidum and MT. This finding may highlight a correlation between awareness of vaccination programmes, high vaccination compliance and comprehensive understanding of infectious disease transmission, suggesting a link between vaccination education and broader knowledge of infectious diseases. In the literature, we can already find a possible association between vaccine hesitancy and knowledge of vaccination [30].

The data also suggest that improved education about immunisation and occupational infections, particularly during the student years (e.g., during medical training), may play a critical role in preventing future problems with vaccine adherence. Previous studies have shown that students’ risk perceptions are low at the beginning of the study period, as evidenced by Ginji et al.’s study in relation to measles [31]. Healthcare professionals who understand the complexities of infectious disease transmission are more likely to adopt preventive measures such as vaccination, both for personal protection and for public health. Vaccine hesitancy, particularly among HCWs, is an ongoing problem that could be reduced by strengthening the link between disease prevention (through vaccination) and occupational health risks [32]. It has been argued that vaccine hesitancy should be addressed early in medical training to protect future health workers, preserve essential health services and reduce the risk of further pandemics [33].

The fact that vaccine hesitancy among health workers remains a persistent problem, as highlighted by previous studies in Albania, has important implications for public health strategies; it is fundamental to assess students’ attitudes to vaccination [34]. HCWs are role models and play a key role in promoting vaccination to the general population.

Statistical analysis using the chi-squared test revealed significant associations between the knowledge of HCV PEP measures (item 10C) and several other variables.

The association between participants’ knowledge of the transmission of Treponema pallidum and the misconception of an HCV prophylaxis regimen (which does not currently exist) may indicate a wider gap in understanding of the routes of disease transmission and available preventive measures. Confusion may arise from the fact that both HCV and Treponema Pallidum can be transmitted by blood or parenterally. This highlights the need for more focused training on the specifics of post-exposure protocols and the non-availability of certain prophylactics, such as the HCV vaccine.

The second significant correlation is related to the question of occupational exposure training for students. Participants who felt that occupational exposure training was inadequate were more likely to believe in the availability of an HCV vaccine for post-exposure prophylaxis. This may reflect inadequacies in curricula or clinical training regarding HCV risks and appropriate post-exposure protocols, highlighting the need to improve medical training in infectious diseases.

The third significant correlation relates to whether risk can influence future career choices, suggesting that those who believe in the existence of an HCV vaccine are also more likely to consider occupational exposure risks in their future career decisions. This suggests that misinformation or gaps in knowledge about available preventive measures could influence participants’ career choices, possibly leading them away from specialties with a higher risk of exposure due to unfounded fears or misunderstandings about protective strategies.

The last correlation, related to the question of whether they ever refused to care for patients for fear of infection, indicates a deeper issue of fear and uncertainty when dealing with infectious patients, particularly if HCW students are uncertain about available prophylaxis options or do not understand post-exposure protocols. This may contribute to stigmatisation of infectious patients and reluctance to engage in certain medical practices, further highlighting the need for comprehensive training on occupational exposure risks and protection methods to reduce fear-based refusals to treat.

The small sample size, unevenly distributed in gender (F = 80.9%), is certainly a limitation of the study, but the 100% response rate reinforces the validity of the results obtained.

## 5. Conclusions

In our study, we found that a small (not negligible) percentage of medical students have little knowledge of occupational infections and biohazards.

Furthermore, although the majority of students recognise a moral obligation to care for infectious patients, there are cases of refusal by some students. Statistical analysis shows a positive correlation between vaccination awareness and knowledge of infectious diseases, with greater vaccination compliance among those with a better understanding of biological risks. This highlights the importance of accurate information to prevent problems with vaccine adherence. The findings indicate that pre-clinical training programmes need to be strengthened to ensure better preparation of students before they enter the hospital environment. Also crucial are information programmes regarding vaccinations to prevent adherence and hesitation problems.

Further investigations are needed to confirm and extend these findings, and we hope that the questionnaire we used can be reproduced in further studies to extend the research.

## Figures and Tables

**Figure 1 diseases-13-00011-f001:**
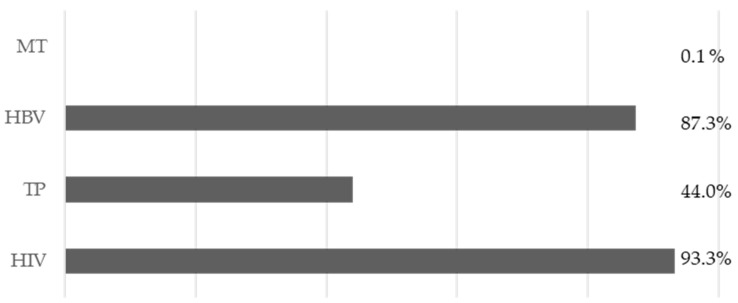
Answers to item 7, “Which of the following pathogenic microorganisms can be transmitted by blood?”

**Figure 2 diseases-13-00011-f002:**
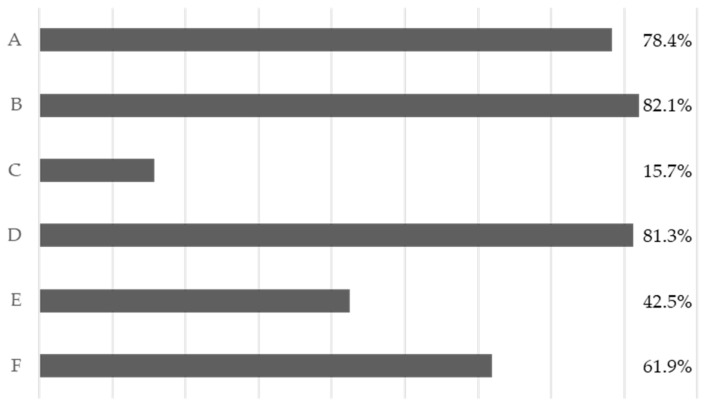
Adherence to the various forms of risk prevention for infectious agents. A: Wearing a mask and working coats. B: Wearing gloves during procedures and patient-care activities. C: Wearing protective eyewear during procedures and patient-care activities. D: Removing gloves and washing hands immediately after operation and correctly antisepticising hands. E: Wearing two pairs of gloves when there are wounds on hands. F: Wearing gloves to clear away instruments and medical waste.

**Table 1 diseases-13-00011-t001:** Characteristics of study population.

		n	%
Total		134	
Gender	Male	26	19.4
	Female	108	80.6
Educational Grade	Undergraduate	131	97.8
	Graduate	3	2.2
Immunisation for HBV		123	91.8

**Table 2 diseases-13-00011-t002:** Cross-tab item 34.

		Is the Flu Vaccine Strongly Recommended for Healthcare Workers?				
		Yes	%	No	%	*p* Value
Gender	M	20	14.9	6	4.5	Ns
	F	80	59.7	28	20.9	
How familiar are you with the problem of occupational exposure to biological agents?	A lot	66	49.3	20	14.9	Ns
	Just a little	34	25.4	14	10.4	
Can HIV be transmitted by blood?	Yes	94	70.1	31	23.1	Ns
	No	6	4.5	3	2.2	
Can Treponema Pallidum be transmitted by blood?	Yes	55	41.0	4	3.0	<0.05
	No	45	33.6	30	22.4	
Can HBV be transmitted by blood?	Yes	89	66.4	28	20.9	Ns
	No	11	8.2	6	4.5	
Tuberculosis is transmitted through respiratory droplets	True	98	73.1	28	20.9	<0.05
	False	2	1.5	6	4.5	
Have you ever suffered a biological injury during your training?	Yes	2	1.5	0	0.0	Ns
	No	98	73.1	34	25.4	
Have you received training on occupational exposure to biological agents and on risk prevention and management measures?	Yes	76	56.7	24	17.9	Ns
	No	24	17.9	10	7.5	
Do you think that training on occupational exposure for medical students is sufficient?	Yes	20	14.9	14	10.4	Ns
	No	54	40.3	46	34.3	
Do you think medical students have a moral obligation and responsibility to treat infectious patients?	Yes	80	59.7	29	21.6	Ns
	No	20	14.9	5	3.7	
Have you ever refused to treat infectious patients because of fear of transmission?	Yes	5	3.7	0	0.0	Ns
	No	95	70.9	34	25.4	
After learning about the risks of occupational exposure to biological agents do you think the risk will influence your future career choices?	Yes	63	47.0	18	13.4	Ns
	No	55	41.0	16	11.9	

**Table 3 diseases-13-00011-t003:** Cross-tab item 35.

		Is the Measles Vaccine Strongly Recommended for Healthcare Workers?				
		Yes	%	No	%	*p* Value
Gender	M	18	13.4	8	6.0	Ns
	F	68	50.7	40	29.9	
How familiar are you with the problem of occupational exposure to biological agents?	A lot	53	39.6	33	24.6	Ns
	Just a little	33	24.6	15	11.2	
Can HIV be transmitted by blood?	Yes	79	59.0	46	34.3	Ns
	No	7	5.2	2	1.5	
Can Treponema Pallidum be transmitted by blood?	Yes	46	34.3	13	9.7	<0.05
	No	40	29.9	35	26.1	
Can HBV be transmitted by blood?	Yes	75	56.0	42	31.3	Ns
	No	11	8.2	6	4.5	
Tuberculosis is transmitted through respiratory droplets	True	84	62.7	42	31.3	<0.05
	False	2	1.5	6	4.5	
Have you ever suffered a biological injury during your training?	Yes	1	0.7	1	0.7	Ns
	No	85	63.4	47	35.1	
Have you received training on occupational exposure to biological agents and on risk prevention and management measures?	Yes	60	44.8	40	29.9	Ns
	No	26	19.4	8	6.0	
Do you think that training on occupational exposure for medical students is sufficient?	Yes	40	29.9	20	14.9	Ns
	No	74	55.2	28	20.9	
Do you think medical students have a moral obligation and responsibility to treat infectious patients?	Yes	69	51.5	40	29.9	Ns
	No	17	12.7	8	6.0	
Have you ever refused to treat infectious patients because of fear of transmission?	Yes	5	3.7	0	0.0	Ns
	No	81	60.4	48	35.8	
After learning about the risks of occupational exposure to biological agents do you think the risk will influence your future career choices?	Yes	39	29.1	24	17.9	Ns
	No	47	35.1	24	17.9	

**Table 4 diseases-13-00011-t004:** Cross-tab item 10C. * PEP: post-exposure prophylaxis.

		Is There an Immediate PEP * Regimen (HCV Vaccine) to Prevent Infection After Occupational Exposure to HCV?				
		Yes	%	No	%	*p* Value
Gender	M	8	6.0	18	13.4	Ns
	F	28	20.9	80	59.7	
How familiar are you with the problem of occupational exposure to biological agents?	A lot	23	17.2	63	47.0	Ns
	Just a little	13	9.7	35	26.1	
Can HIV be transmitted by blood?	Yes	35	26.1	90	67.2	Ns
	No	1	0.7	8	6.0	
Can Treponema Pallidum be transmitted by blood?	Yes	21	15.7	38	28.4	<0.05
	No	15	11.2	60	44.8	
Can HBV be transmitted by blood?	Yes	34	25.4	83	61.9	Ns
	No	2	1.5	15	11.2	
Tuberculosis is transmitted through respiratory droplets	True	33	24.6	93	69.4	Ns
	False	3	2.2	5	3.7	
Have you ever suffered a biological injury during your training?	Yes	0	0	2	1.5	Ns
	No	36	26.9	96	71.6	
Have you received training on occupational exposure to biological agents and on risk prevention and management measures?	Yes	27	20.1	73	54.5	Ns
	No	9	6.7	25	18.7	
Do you think that training on occupational exposure for medical students is sufficient?	Yes	23	17.2	37	27.6	<0.05
	No	13	9.7	61	45.5	
Do you think medical students have a moral obligation and responsibility to treat infectious patients?	Yes	30	22.4	79	59.0	Ns
	No	6	4.5	19	14.2	
Have you ever refused to treat infectious patients because of fear of transmission?	Yes	4	3.0	1	0.7	<0.05
	No	32	23.9	97	72.4	
After learning about the risks of occupational exposure to biological agents do you think the risk will influence your future career choices?	Yes	11	8.2	52	38.8	<0.05
	No	25	18.7	46	34.3	

## Data Availability

Datasets used and/or analysed during the current study are available from the corresponding author on reasonable request.

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
