# Peer review of "A Survey of the Knowledge, Attitudes and Practices of a Sample of Albanian Medical Students in Relation to Occupational Exposure to Biological Agents"

_diseases, 2025, doi:10.3390/diseases13010011_

Round 1
Reviewer 1 Report
Comments and Suggestions for Authors
This research addresses an important problem related to blood born transmissible infections. This is a problem that HCWs meet almost daily in the process of their service.
The however focused more on HBV and HIV and did not consider significantly HCV. Instead went on to look at TB and measles. This is an important question to be addressed.
Otherwise it is well written
Reviewer 2 Report
Comments and Suggestions for Authors
Introduction – the introduction requires supplementation, it is not complete;
1. Lines 42-45 – what is the prevalence of these infections (and morbidity trends) in Albania?
2. How many injuries are there in Albania, but also in other countries of Central and Eastern Europe (please compare to the situation of similar countries, not distant ones); the importance of prevention can be described (e.g. in addition to vaccinations, also the impact / importance of using so-called safe medical equipment in reducing the risk of injury)
3. What is the vaccination program in Albania, what are the mandatory vaccinations (are they?), is vaccination against HBV mandatory for healthcare workers (pupils and students of medical schools?)
4. What are the gaps in other studies
5. And finally, what new things do the authors contribute to the knowledge of exposure to infectious material with their study?
Materials and methods
The topic of the article covers issues related to occupational exposure to blood-borne pathogens; therefore, please explain the purpose of adding the questions “about knowledge of airborne transmission of Mycobacterium tuberculosis and about influenza and measles vaccinations,” – this issue is not directly related to the topic and purpose of the work.
Characteristic of the study group
What fields of study were the students from? From which year of studies? Why did the authors take so few variables (affecting knowledge and attitudes) into account? (especially since the majority were women)
Results – the results should be presented not only as percentages, but also as numbers N
Tables 2 and 3 - I do not understand why the authors checked the relationship between the recommendations for vaccination against measles and influenza and the variables given (e.g. knowledge about HBV transmission) - what is the point of this? What relationship did the authors have in mind? The choice of variables (vaccination against measles and influenza) is also inconsistent with the title of the manuscript or the objectives of the work. I believe that the authors should have performed other statistical analyses.
Table no. 3 is definitely more correct, but it would also be better to check the knowledge about post-exposure prophylaxis against HCV, e.g. with training in this area, (and they do not know whether tuberculosis is transmitted through the air and HBV through the blood); this is incomprehensible to me.
The discussion should be based on the results, not repeat them, and the authors should discuss only the obtained results, and not discuss what they did not include in the results.
The conclusions should correspond to the assumed research objectives.
The references require supplementation.
The language of the manuscript requires verification.
Reviewer 3 Report
Comments and Suggestions for Authors
To the authors
The subject of the training and degree of knowledge of medical students is a subject that is still topical. It is all the more important as it conditions, on the one hand, the safety of health professionals and patients, and on the other hand, the quality and adequacy of the care provided by the staff.
The questionnaire : The study was carried out using a questionnaire. In fact, the questionnaire is taken from the publication by Wu L et al. 2016 (ref.14). The article itself is presented/constructed according to a very similar structure. It should be noted, however, that in the basic Chinese study, the validation of the questionnaire was carried out in dentistry students (not in medicine). In practice, one may indeed wonder whether to what extend the occupational risks incurred by a doctor or by a dentist are different? But we also suspect that if the basic knowledge may be common, differences occur depending on the medical specialty and the gestures related to its practice (e.g. surgeon versus general practitioner, ...).
The study group : As shown in Table 1, women make up 80.6% of the group. How do we explain this majority of women? Is it common among medical students in Albania? In Balkan countries ?
The classification into "undergraduate" versus "graduate" (table 1 line 89) shows a very large majority of "undergraduate" (97.8%). We can therefore assume that these students are very young. It should be noted that age was not included in the parameters considered during the cross-analyses: probably due to too much homogeneity?
But, in the same context, one may wonder whether a very young student - who therefore has neither practice nor experience yet - is able to make an objective judgment on the quality of the training he has just received in a field he did not know? Isn't it rather a subjective sensation linked to the fear of risk?
It would have been interesting, by way of comparison, for the authors to give some information on the results obtained in a series of students from local schools. In this study, there is no control series and this is lacking.
As expressed here, the results conclude/imply that the quality of education in Albania needs to be improved, with Albania representing one of the Balkan steps.
At first, this observation appears to be a rather pejorative value judgment. But in a second step, especially after consulting reference 22 (Gjini E et al. 2022, ref. 22), it is clear that the attitude of the Albanian population towards the vaccination of children needs to be improved. And that to do this, medical and nursing staff must have a solid knowledge base to convince the population of the public health benefits of vaccination. This is a very crucial question since, as the authors say : “HCWs are role models and play a key role in promoting vaccination to 202 the general population.”
Alessandro's study (ref 15) concludes that the score obtained by nurses is better than that of doctors. He sums up his questioning with a challenging reflection that calls for questioning the teaching methods to be used. It is formulated as follows : "what is working better in nursing than in medical education in order to implement and validate new teaching approaches.".
This is a very crucial question since, as the authors say : “HCWs are role models and play a key role in promoting vaccination to the general population.”
Bibiography : The bibliography is well stocked. However, some references to the same themes date from quite different years. Given the speed with which the means of communication, the didactic tools, the progress of the doctor and mentalities are evolving, we can ask ourselves the question of their alignment in relation to the same subject.
Example: lines 49-53 : ref 6 (1998), ref 7 (2023), ref 8 (2016), ref 9 (2007). Shouldn't we take the precaution of specifying that in medicine, as in many other fields, the integration of newly acquired knowledge takes time and the application of the new safety instructions too
Formal remarks:
Line 89 Table 4: The term PEP that first appears in the text at this location should be explained here.
Line 95: Table 1: MT for Mycobacterium tuberculosis appears in the histogram but the term "MT" has not yet been defined in the text; it is only in point 7 of the questionnaire (whereas TP is defined in the introduction line 37).
Line 162-163 : “Unfortunately, the results showed that a significant percentage of the students had a significant lack of knowledge about healthcare-associated infections.”.
The authors therefore conclude that a significant proportion of students have a lack of knowledge about infectious diseases. At what percentage of correct answers is the overall result of such a questionnaire considered "satisfactory"?
The solution proposed by the authors (support and supervision) is obviously the right one; it helps to fill in the gaps.
Line 172-173 : “Although 75% of students had received training on occupational infections and biohazards, only 45% felt that this was sufficient.”
On what basis can 45% of these young students (i.e. almost one in 2) conclude that what they have been taught is insufficient?
Line 178-179: "3.6% say they have not treated a sick patient for fear of infectious disease". The percentage indicated refers to the series studied for a point that is ethical, of a mentality; We could better appreciate its importance if we had a point of comparison.
Line 188-189: the impact of the attitude towards vaccination is obviously fundamental, as also indicated in ref 22.
Line 280 : Occupational exposure questionnaire for mental students 280 (à dental).
_________________________________________________________________________________
